# Mitochondrial Dynamics in Neurodegenerative Diseases: Unraveling the Role of Fusion and Fission Processes

**DOI:** 10.3390/ijms241713033

**Published:** 2023-08-22

**Authors:** Hubert Grel, Damian Woznica, Katarzyna Ratajczak, Ewelina Kalwarczyk, Julia Anchimowicz, Weronika Switlik, Piotr Olejnik, Piotr Zielonka, Magdalena Stobiecka, Slawomir Jakiela

**Affiliations:** 1Department of Physics and Biophysics, Institute of Biology, Warsaw University of Life Sciences, 02-787 Warsaw, Poland; 2Department of Biochemistry and Microbiology, Institute of Biology, Warsaw University of Life Sciences, 02-787 Warsaw, Poland

**Keywords:** neurodegenerative disease, mitochondrial fusion, mitochondrial fission, mitochondrial dynamics, potential drugs

## Abstract

Neurodegenerative diseases (NDs) are a diverse group of disorders characterized by the progressive degeneration and death of neurons, leading to a range of neurological symptoms. Despite the heterogeneity of these conditions, a common denominator is the implication of mitochondrial dysfunction in their pathogenesis. Mitochondria play a crucial role in creating biomolecules, providing energy through adenosine triphosphate (ATP) generated by oxidative phosphorylation (OXPHOS), and producing reactive oxygen species (ROS). When they’re not functioning correctly, becoming fragmented and losing their membrane potential, they contribute to these diseases. In this review, we explore how mitochondria fuse and undergo fission, especially in the context of NDs. We discuss the genetic and protein mutations linked to these diseases and how they impact mitochondrial dynamics. We also look at the key regulatory proteins in fusion (MFN1, MFN2, and OPA1) and fission (DRP1 and FIS1), including their post-translational modifications. Furthermore, we highlight potential drugs that can influence mitochondrial dynamics. By unpacking these complex processes, we aim to direct research towards treatments that can improve life quality for people with these challenging conditions.

## 1. Introduction

NDs are complex, influenced by a web of interactions among various brain cell types and systems. Elements like neuroinflammation, synaptic dysfunction, impaired protein clearance, and genetic predispositions, all play a part in ND development [1,2]. To untangle these interactions and understand NDs better, ongoing research is critical. Current studies are delving into the roles of the immune system, neuroinflammation, genetic and epigenetic factors, and utilizing advanced imaging techniques for early detection.

Advanced stages of NDs cause significant changes to neural connections, cellular activity, and can even lead to nerve cell death. This is particularly noticeable in brain regions like the entorhinal cortex and hippocampus. Although there has been substantial progress in understanding NDs, the exact triggers and the manner in which they progress are not entirely clear. The aggregation of proteins and mitochondrial dysfunction are identified as key factors in the development of these diseases [3].

Despite the strides made in targeted therapies, which include cell therapies and new drugs, repairing the damage done to the nervous system is a formidable task [4]. NDs rank high among deadly diseases in the United States, placing them among the significant challenges in modern medicine [5]. As a result, extensive research efforts focus on understanding and addressing the multifactorial nature of NDs.

Mitochondria, often referred to as the cell’s power plants, play a crucial role in NDs such as Alzheimer’s and Parkinson’s [6,7]. They generate energy via ATP, produce reactive oxygen species (ROS) through OXPHOS, regulate calcium equilibrium, and initiate pathways that lead to cell death [8,9]. However, under conditions of stress, aging, or in the presence of NDs, these nerve cell mitochondria can fragment and malfunction.

Mitochondrial dysfunction, characterized by a loss of membrane potential, results in an energy supply deficit and oxidative stress [6,10]. The degenerative processes in nerve cells start long before the onset of mental symptoms, and once these symptoms become apparent, the disease progresses rapidly, having a significant impact on patients and their families [10,11]. Currently, the lack of reliable biomarkers makes early diagnosis challenging [12,13].

Mitochondria are not static but rather dynamic organelles, continuously undergoing fusion and fission, processes that alter their morphology and function. Several mitochondrial dysfunctions associated with NDs include increased ROS production, α-synuclein aggregation impacting mitochondrial transport, abnormal morphology and distribution affecting ATP delivery, reduced mitochondrial biogenesis, increased mitophagy for degrading damaged mitochondria, and disrupted fusion and fission processes essential for mitochondrial dynamics and function. The role of mitochondrial dysfunction dynamics might not be fully appreciated yet [14,15].

In this comprehensive review, the authors offer a detailed analysis focusing on the fusion and fission processes in mitochondria within the context of NDs. The intention is to shed light on the intricate interactions underpinning ND development. The ultimate goal is to pave the way for targeted interventions and therapeutic strategies to address the urgent challenges NDs pose in modern medicine, and ultimately improve patients’ quality of life.

## 2. Neurodegenerative Diseases—Gene Mutations

NDs are a broad category that includes conditions like Alzheimer’s disease (AD), Parkinson’s disease (PD), Amyotrophic Lateral Sclerosis (ALS), Huntington’s disease (HD), Multiple Sclerosis (MS), Frontotemporal Dementia (FTD), Prion diseases such as Creutzfeldt–Jakob disease (CJD), and other less common disorders. Studying these conditions has provided us a deeper understanding of the genes and proteins involved in their development.

AD is associated with genetic mutations in key genes, including Amyloid-β Precursor Protein (APP), Presenilin 1 (PS1), and Presenilin 2 (PS2), which are strongly linked to familial forms of the disease [16]. Moreover, the apolipoprotein E (APOE) gene is a significant genetic risk factor for sporadic AD, impacting both susceptibility to and progression of the disease [17].

PD is associated with several genes in familial cases. These include mutations in PARK2, PINK1, PARK7 (DJ-1), and LRRK2 [18]. Mutations in the GBA gene (glucocerebrosidase) are also viewed as significant genetic risk factors for PD development, providing more insight into the disease’s complex genetic basis [18,19].

ALS is associated with a variety of genetic mutations. Genes like superoxide dismutase 1 (SOD1), C9orf72, TAR DNA-binding protein (TARDBP), and Fused in Sarcoma (FUS) have been linked to both familial and sporadic cases of ALS, indicating their involvement in the disease’s development [20,21,22].

HD is caused by an abnormal expansion of CAG repeats in the huntingtin gene (HTT) [23]. This critically influences the age of onset and disease severity. Fully understanding the genetic basis of HD is crucial for clarifying its complex molecular mechanisms and creating effective therapeutic strategies [24].

MS has a complex etiology that involves a mix of genetic and environmental factors. Genetic risk elements tied to MS include variants in genes like HLA-DRB1, IL7R, and IL2RA [25,26]. However, the genetic part of MS is complicated, featuring multiple genes and complex gene-environment interactions. A complete understanding is needed to unravel MS’s underlying mechanisms and develop personalized treatment approaches [26,27].

FTD is associated with several genetic risk factors. Mutations in the microtubule-associated protein tau (MAPT) gene, coding for the tau protein, have been detected in certain FTD cases. These mutations can upset tau’s normal function and lead to abnormal protein aggregations in the brain [28]. Mutations in other genes, like progranulin (GRN) and C9orf72, are also connected with FTD, impacting the production or processing of specific proteins involved in cellular processes [29,30]. This leads to protein aggregation and neuronal damage.

Prion diseases, including CJD, are marked by the buildup of misfolded prion proteins in the brain. Sometimes, prion diseases can be caused by inherited genetic mutations in the PRNP gene, which codes for the prion protein [31,32,33]. These mutations can alter the protein’s structure, encouraging misfolding and aggregation. The abnormal prion protein can then cause the normal prion proteins to convert into the disease-associated form, triggering the spread of pathology and neurodegeneration [31,32,33].

Recent research has explored alterations in protein expression linked with mitochondrial function, including OXPHOS, and key elements like PGC-1α, MnSOD, CypD, HIF-1α [34,35,36,37]. Also studied are proteins governing mitochondrial dynamics, such as DRP1, MFN1, MFN2, OPA1, and FIS1 [38,39,40]. Dysregulation or mutations coding these proteins have emerged as key factors in the context of neurodegenerative disorders, underlining their importance in the pathogenesis of these conditions [38,39,40].

## 3. Regulatory Proteins in Mitochondrial Fusion and Fission

Mitochondrial dynamics is orchestrated by key proteins, dynamin-related protein 1 (DRP1) and fission protein 1 (FIS1) direct the fission event, whereas mitofusins (MFNs) and optic atrophy 1 (OPA1) supervise the fusion process.

### 3.1. Mitochondrial Fission Protein 1 (FIS1)

FIS1 holds a pivotal role in the regulation of mitochondrial dynamics, primarily situated in the outer mitochondrial membrane. Ranging from approximately 17 to 20 kiloDaltons, FIS1 is composed of a singular transmembrane domain which acts as an anchor to the outer mitochondrial membrane (Figure 1) [41]. The expression and functional activity of FIS1 are under stringent regulation to ensure the preservation of optimal mitochondrial dynamics, with its modulation controlled by an array of cellular factors and signaling pathways [42].

FIS1 facilitates mitochondrial fission by enlisting the participation of proteins essential to this process, among which is the DRP1, a key protagonist in mitochondrial fission. Furthermore, FIS1 engages in interactions with other key players in mitochondrial dynamics, such as MFNs and OPA1, to orchestrate a balance between mitochondrial fission and fusion events [43].

While research on FIS1 is somewhat limited, emerging evidence points to alterations in its expression and function in NDs like AD and PD. Dysregulation of FIS1 has been implicated in mitochondrial fragmentation and dysfunction, adding to the pathological processes underlying these disorders [42].

### 3.2. Dynamin-Related Protein 1 (DRP1)

DRP1, interchangeably referenced as dynamin-related protein 1-like (DLP1) or dynamin-1-like protein (DNM1L), has a vital function in the modulation of mitochondrial dynamics, specifically through the orchestration of mitochondrial fission. DRP1 typically resides in the cytoplasm, but it migrates to the outer mitochondrial membrane under conditions of activation. The transition of DRP1 to the mitochondria is controlled by specific adaptor proteins and is subject to a range of post-translational modifications [44,45].

Composed of 736 amino acids (according to UniProt), DRP1 harbors four distinct functional domains: a highly conserved N-terminal GTPase domain, a middle domain, a variable domain, and a C-terminal GTPase effector domain (GED) (Figure 1). Deletion of any of these three domains (GED, GTPase, or middle domain) impairs the mitochondrial function of DRP1. The activity and subcellular distribution of DRP1 are fine-tuned by various post-translational modifications, including phosphorylation, sumoylation, ubiquitination, and S-nitrosylation, which, in turn, influence broader mitochondrial dynamics [46].

The GTPase domain, critical for GTP binding and hydrolysis, triggers conformational changes that are essential for mitochondrial constriction and subsequent division, also known as fission. The interaction of the GTPase domain with GTP activates DRP1 oligomerization and initiates its recruitment to the outer mitochondrial membrane. There, it assembles into a constriction site, which instigates mitochondrial division. This mechanism is critical for maintaining mitochondrial quality control, facilitating mitochondrial turnover via mitophagy, and assuring proper mitochondrial distribution within cells. The dysregulation of DRP1 and any resulting aberrations in mitochondrial fission have been linked to neurodegenerative disorders and cancers [47,48].

Mutations or dysregulation in DRP1 have been associated with several neurodegenerative disorders, like AD, PD, and HD. Disturbances in DRP1 function upset the delicate balance of mitochondrial fission and fusion processes, leading to mitochondrial dysfunction and subsequent neuronal damage, exacerbating the pathogenesis of these severe disorders [49].

### 3.3. Mitofusins (MFNs)

MFNs represent integral mitochondrial proteins that aid in the process of mitochondrial fusion. Predominantly situated in the outer mitochondrial membrane, two primary isoforms exist: MFN1 and MFN2. Composed of approximately 741 amino acids, MFN1 includes two transmembrane domains, a coiled-coil domain, a GTPase domain, and a helix bundle domain formed by multiple helices dispersed across MFN2 sequence (Figure 1). Conversely, MFN2, with a length of around 757 amino acids, exhibits structural parallels with MFN1. Both isoforms undergo a variety of post-translational modifications, such as phosphorylation and ubiquitination, which serve to impact their activity and localization within the cell [50].

MFN1 and MFN2 function synergistically to enable correct mitochondrial fusion, attaching the outer mitochondrial membranes and promoting the fusion of the inner mitochondrial membranes. The formation of homo- and hetero-oligomers aids in the efficient communication between mitochondria and maintenance of mitochondrial integrity, connectivity, and network formation [51].

The dysregulation of MFNs has been implicated in a diverse range of diseases, encompassing neurodegenerative disorders, metabolic disorders, and cardiovascular diseases [52]. Mutations or disruptions in mitofusin proteins can lead to an imbalance in mitochondrial dynamics and have been associated with several human diseases [53]. For example, mutations in MFN2 cause Charcot–Marie–Tooth disease type 2A, as well as ND affecting the peripheral nerves [54].

### 3.4. Optic Atrophy Protein 1 (OPA1)

OPA1 serves as a pivotal regulator of mitochondrial fusion, cristae remodeling, and overall mitochondrial functionality. Primarily located in the inner mitochondrial membrane, OPA1 is subject to extensive alternative splicing, thereby giving rise to distinct isoforms, each characterized by unique functionalities. The activity and cellular localization of OPA1 are modulated through proteolytic processing and diverse post-translational modifications such as phosphorylation and ubiquitination [55].

OPA1 is a large GTPase protein that undergoes extensive alternative splicing, resulting in multiple isoforms. These isoforms, including short forms (S-OPA1) and long forms (L-OPA1), vary in length and domain composition, each serving distinct roles and functions within the mitochondria. The activity of OPA1 is regulated by various factors, including proteolytic processing and post-translational modifications. Proteases such as OMA1 and YME1L mediate proteolytic processing, generating distinct isoforms of OPA1 with specific functions. Additionally, OPA1 undergoes phosphorylation and ubiquitination, which impact its stability and function [55].

OPA1 orchestrates the fusion of inner mitochondrial membranes, preserving mitochondrial structure, functionality, and cellular homeostasis. Its role in guiding mitochondrial fusion underpins the interconnectedness of the mitochondrial network and promotes the exchange of mitochondrial components, contributing to the maintenance of mitochondrial integrity and overall function [55].

In the outer mitochondrial membrane, OPA1 interacts with mitofusins, MFN1, and MFN2. This interaction facilitates the formation of homo- and hetero-oligomers, enabling coordinated fusion events. Perturbations in OPA1 function can result in compromised mitochondrial fusion, disruption of the cristae structure, and other mitochondrial abnormalities, which have been implicated in the etiology of various neurodegenerative disorders [55].

Mutations in OPA1 are primarily associated with autosomal dominant optic atrophy (ADOA), a neurodegenerative disorder characterized by vision loss. OPA1 mutations disrupt mitochondrial fusion, resulting in impaired mitochondrial morphology and function within retinal ganglion cells. Understanding the mechanisms underlying OPA1-mediated disruption in mitochondrial dynamics offers great potential to clarify the pathogenesis of ADOA and find potential therapeutic strategies [56].

## 4. Mitochondrial Fusion and Fission: Key Tenets of Mitochondrial Quality Control

Mitochondrial fusion and fission are vital operations that underpin the maintenance of mitochondrial quality control [57]. These mechanisms enable the segregation of damaged mitochondria and facilitate the equilibration of mitochondrial components. The morphological adjustments within the mitochondrial network exert a profound influence on their biochemical attributes. For instance, an uptick in mitochondrial divisions yields smaller, and occasionally degenerate, mitochondria that may be devoid of mitochondrial DNA (mtDNA) [58]. Despite potential degeneration, their compact size and altered form aid in their navigation through the complex cytoskeletal protein networks to remote regions of the cell, such as the dendrites of Purkinje neurons [57,59]. Neurons, with their high energy demands, heavily depend on healthy mitochondria to uphold proper functionality. Consequently, defects in mitochondrial maintenance mechanisms can trigger the accumulation of impaired mitochondria and subsequent neurodegeneration.

Twig and colleagues underscored the indispensability of mitochondrial fission as a crucial process for the targeted elimination of damaged mitochondria through autophagy [60]. This observation lends substantial credence to the notion that mitochondrial fission and fusion collectively stand as pivotal mechanisms governing mitochondrial quality control. Similarly, Song et al. shed light on the intricate interconnectedness characterizing mitochondrial dynamics, mitophagy, and the ubiquitin-proteasome system (UPS), which collectively contribute to maintaining cardiovascular homeostasis [61]. Their work underscores the overarching influence exerted by mitochondrial dynamics, which encompass fission and fusion, over the regulation of mitophagy.

Contrary to contrasting viewpoints, Narendra and colleagues presented compelling evidence substantiating the capacity for mitophagy to occur independently of mitochondrial dynamics [62]. They highlight the pivotal role played by the E3 ubiquitin ligase Parkin in recruiting impaired mitochondria, thereby instigating the autophagic process. This finding posits the hypothesis that mitophagy initiation can indeed proceed autonomously of the conventional processes of mitochondrial fission and fusion.

However, a broader perspective was introduced by Pickles et al., who delved into the realm of mitochondrial quality control mechanisms [63]. Their investigation underscores the evolution of multifaceted quality control systems within mitochondria, ensuring their functional presence. This comprehensive framework encompasses not only mitophagy but also extends to encompass additional processes, including biogenesis and protein turnover. This holistic understanding indicates that the enhancement of mitochondrial quality is contingent upon more than just the modulation of mitochondrial movement and interactions within cells; it encompasses a spectrum of interrelated mechanisms.

Elaborating on this concept, Wang et al. illuminated the synergistic interplay and interconnectedness inherent in the mechanisms governing mitochondrial quality control across diverse cellular pathways [64]. They propose that any disruption introduced to a single cellular pathway can induce cascading effects throughout the cellular network, thereby exerting an impact on fundamental processes like mitochondrial fusion and fission. This viewpoint alludes to the idea that, while the pivotal roles of mitochondrial fusion and fission in shaping mitochondrial quality control are undeniable, they might not exclusively determine its regulation. Instead, other contributory mechanisms, such as mitochondrial autophagy, emerge as equally significant players in the holistic preservation of overall mitochondrial health.

Importantly, fragmented mitochondria undertake important roles in thermogenesis within brown adipose tissue, enhancing the efficacy of mitophagy and ultimately modulating apoptosis [65]. These processes are crucial for cellular function, as mitophagy allows cells to purge damaged mitochondria, consequently reducing free radical production [48,59]. Moreover, under high-stress conditions, an increase in apoptosis serves a beneficial role in eliminating damaged cells, suggesting that mitochondria play an indirect role in intercellular signaling to notify the cellular environment of external and internal threats [59].

The process of mitochondrial fusion is also of considerable significance, especially in energy-demanding cells like neurons and muscle cells [59,66]. Mitochondrial fusion facilitates the exchange of metabolites, enzymes, and mitochondrial gene expression products throughout the mitochondrial compartment. This becomes notably relevant during periods of heightened cellular energy demand. The ensuing elongation of mitochondria amplifies the efficiency of OXPHOS, the primary mechanism of adenosine 5′-triphosphate (ATP) production [67]. These insights indicate that the dynamic remodeling of mitochondria, impacting their structure, plasticity, and functions, significantly bolsters ATP production [66]. Furthermore, the morphological adaptations of mitochondria enable their efficient movement to cellular regions with high ATP requirements [68,69].

In neurodegenerative disorders, the processes of fusion and fission of mitochondria in cells are always disturbed, resulting in significant impairment of ATP distribution [59,66] and defective mitophagy [70]. As a consequence, progressive degeneration and death of neurons occur, leading to various neurological symptoms and cognitive decline [3]. Therefore, in the next part of the article, we explore the dynamic machinery of mitochondria in cells.

## 5. Mitochondrial Fission: Two Pathways

According to the latest reports, mitochondrial fission in mammalian cells can occur in two ways—midzone and peripheral (Figure 2). Researchers like Kleele, Rey, and others highlight the significantly different properties of both pathways [71].

The midzone fission takes place in properly functioning cells and serves as an indicator of healthy mitochondria. It occurs without any undesirable changes, such as a reduction in cell membrane polarization or an increase in ROS. This type of division also occurs after cellular proliferation and predicts the fate of the organelles and the entire cells. Both splitting products in this division contain functional copies of the replicating mtDNA, ensuring proper functioning of both organelles. This stands in contrast to peripheral fission, where the mitochondria undergoing division are often damaged or partially degenerate. In such cases, peripheral division is preferred when one pole of the mitochondrion shows a decrease in membrane potential and an increase in ROS, while the rest of the organelle remains functional. After peripheral division, the smaller product lacks replicating mtDNA, suggesting its direction towards mitochondrial death, mitophagy.

Although the mechanics of division in both cases rely on the activity of the same proteins, central division involves the attachment of the endoplasmic reticulum, while peripheral fission relies on anchor points binding to another organelle, the lysosome [71,72,73].

It is imperative to recognize that the processes of fusion and fission are not arbitrary events. Instead, they are meticulously controlled to uphold equilibrium. An interruption in this delicate balance, akin to a dissonant note in a musical composition, has the potential to trigger cellular disharmony and disease. Evident in various neurodegenerative disorders such as Alzheimer’s, Parkinson’s, and Huntington’s diseases [74], the elevation of fission, reduction in fusion, and the presence of fragmented mitochondria imply that mitochondrial fragmentation can indeed function as a disease-state biomarker [75].

## 6. Protein Machinery in Mitochondrial Fission

At the core of mitochondrial fission is DRP1, a cytosolic protein, that initiates the fission process within mitochondria (Figure 3) [66,71]. DRP1 assembles at the division site, forming a ring-like structure that tightly encircles and mechanically separates the organelle. Several molecules in the outer mitochondrial membrane aid in the recruitment of DRP1, including FIS1, MFF, MiD49, and MiD51 [76]. However, the exact contribution of each of these proteins to the fission process is a subject of discussion. For example, while FIS1 is recognized as an important receptor for DRP1, its essential role in peripheral fission is debated, with some studies reporting negligible impact on the division process following FIS1 knockout in mammalian cells [77,78]. Furthermore, FIS1’s involvement in lysosomal tagging suggests a role in peripheral fission [79]. The existence of the MFF protein, absent in yeast, suggests significant evolutionary modifications in mammalian mitochondrial division compared with simpler eukaryotic organisms [80].

Despite advancements, the dynamics of the division process are yet to be fully unveiled. Interestingly, FIS1 protein deficiency does not impede mitochondrial division; however, FIS1 overexpression triggers mitochondrial fragmentation, even in the absence of DRP1 [81,82]. Recent studies propose a dual role for FIS1, functioning as both a fission activator and a fusion inhibitor by modulating the GTPase activity of MFN1, MFN2, and OPA1 proteins involved in the fusion process [60]. These discoveries highlight the intricate interplay among the proteins that regulate the balance between mitochondrial division and fusion [47].

The role of post-translational modifications of proteins engaged in mitochondrial division also warrants attention. Phosphorylation plays a pivotal role in modulating DRP1 dynamics [46]. In the human DRP1 isoform 1, phosphorylation occurs at serines 616 and 637 [46]. The Cdk1/cyclin B signaling pathway, regulated by the cell cycle, drives serine 616 phosphorylation, resulting in mitochondrial fragmentation [83,84]. In contrast, serine 637 phosphorylation is facilitated by cAMP-dependent protein kinase A (PKA), inhibiting DRP1’s GTPase activity and preventing its migration to mitochondria [81]. Importantly, the reversible nature of DRP1 phosphorylation profoundly influences its functions [85]. For instance, calcineurin-mediated dephosphorylation of p-DRP1-S637 recruits DRP1 to mitochondria, promoting mitochondrial fragmentation and synaptic dysfunction [86].

Disruptions in mitochondrial fission have been observed in NDs. In AD, increased DRP1 protein expression is noted in patients [87]. Reducing DRP1 activity has shown potential as a therapeutic approach in AD models [88]. In PD, DRP1-mediated mitochondrial fragmentation is associated with dopaminergic neuronal loss, and inhibiting DRP1 or promoting mitochondrial fusion has demonstrated neuroprotective effects in PD models [89]. Similarly, in HD, abnormal DRP1 activation contributes to mitochondrial fragmentation and dysfunction, leading to neuronal degeneration [49]. Studies have shown that inhibiting the fission machinery through DRP1 or FIS1 knockdown results in the accumulation of oxidized mitochondrial proteins, reduced respiration, and impaired insulin secretion, which are associated with the progression of neurodegenerative diseases [60,90,91,92,93].

## 7. Protein Machinery in Mitochondrial Fusion

Mitochondrial fusion (Figure 3) is a critical cellular process that enhances energy production through efficient OXPHOS [94]. It involves two phases, outer mitochondrial membrane (OMM) fusion, and inner mitochondrial membrane (IMM) fusion, each facilitated by distinct proteins [58]. The process relies on the collaborative actions of two classes of proteins: dynamin-like MFNs and OPA1.

Mitofusins, including MFN1 and MFN2, play essential roles in outer mitochondrial membrane fusion. Removing either of these proteins significantly inhibits fusion, underscoring the necessity of both MFN1 and MFN2 for effective fusion. The GTP binding domain of mitofusins plays a central role in promoting the fusion process [95,96]. Similar to DRP1 in mitochondrial fission, post-translational modifications regulate mitofusins. Phosphorylation, for instance, can lead to the inactivation and breakdown of mitofusin 2 [97]. Additionally, MFN1 phosphorylation by ERK inhibits mitochondrial fusion, while phosphorylated MFN1 stimulates outer mitochondrial membrane permeabilization and apoptosis [98,99].

MFN1 and MFN2 possess a unique transmembrane domain that allows them to penetrate the outer membrane twice, with their N- and C-terminal domains facing the cytosol [100,101]. When two MFNs from this group interact, GTP binding and/or hydrolysis induce conformational changes that enable them to physically bring the external mitochondrial membranes closer, increasing their contact surface area [102]. Although internal membrane fusion can also occur simultaneously, it can be separated from external fusion under specific circumstances. Research indicates that MFN proteins primarily facilitate outer membrane fusion, while OPA1 takes on a dominant role in inner membrane merging [103,104].

OPA1, on the other hand, undergoes proteolytic transformations within mitochondria, generating long (l-OPA1) and short (s-OPA1) forms. The long forms secure themselves to the inner mitochondrial membrane and are essential for efficient fusion, while the short forms promote fission [104,105]. Other modifications, such as ubiquitination, acetylations, and deacetylations, also modulate OPA1’s functions, contributing to the mechanics of mitochondrial fusion [106]. However, challenges remain in distinguishing between modified and unmodified forms of OPA1 using monoclonal antibodies and understanding the proteins’ lifespan within the cell [107].

Several studies have consistently shown that levels of OPA1 and MFNs are significantly reduced in NDs [96,108]. Chen et al. demonstrated that reducing OPA1 levels to 30% of wild-type cells resulted in a 33% reduction in mitochondrial fusion [96]. Similarly, Saita et al. found that depletion of Drp1, a protein involved in mitochondrial fission, led to decreased levels of MFNs and OPA1 [108]. The other studies highlight the significant linkage between parkin-dependent mitophagy and neurodegenerative diseases, including Alzheimer’s and Parkinson’s diseases [109,110,111].

Overall, understanding the cooperation between MFNs and OPA1 in mitochondrial fusion provides essential knowledge for maintaining cellular energy balance and may offer potential targets for therapeutic interventions in conditions related to mitochondrial dysfunction, including NDs [92].

## 8. Potential Therapeutic Agents Modulating Mitochondrial Dynamics

Several biological entities significantly influence the regulation of mitochondrial dynamics, encompassing both fission and fusion. For example, norepinephrine (NE) triggers rapid and complete fragmentation of mitochondria within brown adipose tissue cells [112]. Additionally, in 2022, a novel regulatory protein known as CLUE was discovered to govern the recruitment of DRP1 from the cytosol to the mitochondria [113]. The expression level of CLUE directly determines the degree of mitochondrial remodeling [113].

Inositol, an endogenously synthesized substance, interacts with 5’AMP-activated kinase (AMPK) to curtail its activity, thus inhibiting mitochondrial division [114]. The SOCS6 protein also impacts mitochondrial fragmentation, with diminished expression corresponding to the inhibition of this process [115]. Brain-derived neurotrophic factor (BDNF) is another notable substance, known to enhance the fission process, although further investigations are required to elucidate its exact mechanism [116].

Furthermore, melatonin, a hormone involved in regulating sleep–wake cycles, has been found to modulate mitochondrial dynamics [117,118,119]. Melatonin has been shown to promote mitochondrial fusion and inhibit mitochondrial fission, leading to improved mitochondrial function and protection against oxidative stress [120,121]. In models of NDs, melatonin treatment has been found to reduce mitochondrial fragmentation, enhance mitochondrial function, and protect against neuronal damage [122,123].

Another potential avenue for therapeutic intervention is the regulation of mitochondrial dysfunction-induced cell apoptosis. Herbal medicines have been investigated for their potential to regulate mitochondrial dysfunction and prevent/treat NDs [124,125]. Natural agents derived from herbal medicines have shown beneficial effects in suppressing apoptosis through the regulation of mitochondrial dysfunction [124,125]. These findings provide a foundation for the development of candidate drugs from herbal medicine for NDs [124,125].

Moreover, the regulation of mitochondrial dynamics is not limited to intracellular pathways. External chemical compounds also significantly influence mitochondrial dynamics, either directly or indirectly. Some of these compounds, such as hydrazone M1 [120,126,127,128], BGP-15 [129], echinacoside [130], honokiol [131], and paeonol [132] stimulate mitochondrial fusion. Conversely, julibrazide J13 [133] and all-trans retinoic acid (ATRA) [134] are known to promote mitochondrial fission.

Additionally, sirtuins, a class of proteins involved in regulating cellular metabolism and stress response, have been implicated in mitochondrial quality control and NDs [135]. Modulating sirtuin-mediated mitochondrial quality control through exercise training, calorie restriction, and sirtuin modulators may have therapeutic applications for NDs [135].

Mitochondrial division inhibitors, such as Dynasore [136,137], P110 [138,139], FLZ [140], zinc oxide nanoparticles [141], and Mdivi-1 [142,143], have also been identified. Mdivi-1, whose mode of action is yet to be fully elucidated, is thought to act as a reversible inhibitor of mitochondrial complex I in OXPHOS [144]. Even though it may not be a specific DRP1 inhibitor, subsequent studies have clarified its functionality. Notably, Manczak et al. observed a decrease in DRP1 GTPase activity, resulting in reduced mitochondrial fragmentation and improved cellular function following Mdivi-1 treatment [88]. Ruiz et al. reported that Mdivi-1 inhibits the fission of mitochondria stimulated by NMDA receptor activation, which takes place in excitotoxicity induced by glutamic acid in neuron s [142]. In addition to these effects, Mdivi-1 has been found to exhibit other neuroprotective properties, such as reducing cytosolic Ca^2+^ concentration and calpain activation [142,143,145]. It is noteworthy that Mdivi-1 does not act in the same manner as DRP1 silencing; it independently inhibits mitochondrial fragmentation and cell death in NMDA-induced excitotoxicity [142,146]. These findings highlight the potential of Mdivi-1 as a therapeutic candidate for improving ND outcomes, involving both DRP1-dependent and independent pathways [143].

## 9. Conclusions

In summary, maintaining a delicate balance between mitochondrial fusion and fission processes is vital for cellular homeostasis and optimal neuronal function. Disturbances in these processes, mediated by mutations or changes in the expression of key proteins such as DRP1, FIS1, MFNs, and OPA1, have increasingly been implicated in the pathogenesis of various NDs. Understanding the intricate interplay between these proteins and the molecular and cellular mechanisms underlying mitochondrial dynamics offers valuable insights into the etiology of NDs [15,147]. This knowledge paves the way for the development of targeted therapeutic strategies aimed at restoring or maintaining proper mitochondrial dynamics, potentially providing a novel approach to tackling these debilitating conditions.

Contemporary research also explores potential therapeutics capable of modulating these mitochondrial transformations. Compounds that influence either mitochondrial fusion or fission are promising contenders for novel therapeutic strategies targeting NDs [15,74,148]. For instance, mitochondrial division inhibitors like Mdivi-1 have demonstrated neuroprotective capabilities in conditions induced by excitotoxicity [142,143,149,150].

Given the multifactorial nature of NDs, sustained exploration of the molecular and cellular machinery underlying mitochondrial dynamics is imperative. Interventions aimed at reinstating or sustaining appropriate mitochondrial fusion and fission can unlock novel treatment avenues for these disabling disorders. Such breakthroughs are of profound importance for overcoming the challenges that NDs present in modern medicine, with the overarching goal of enhancing patient quality of life and welfare.

Future research should continue to delve into this promising area, further exploring the role of mitochondrial dynamics in neurodegeneration and how this knowledge can be harnessed for therapeutic gain.

As our comprehension of mitochondrial dynamics and their implications in NDs deepens, ongoing research endeavors and cross-disciplinary collaborations will be vital for deciphering the complex pathophysiology of these diseases and uncovering novel therapeutic targets and strategies [151]. Through these ongoing efforts, we aspire to lay the groundwork for more efficacious treatments and potential cures for NDs, providing a beacon of hope for millions of patients and their families globally.

## Figures and Tables

**Figure 1 ijms-24-13033-f001:**
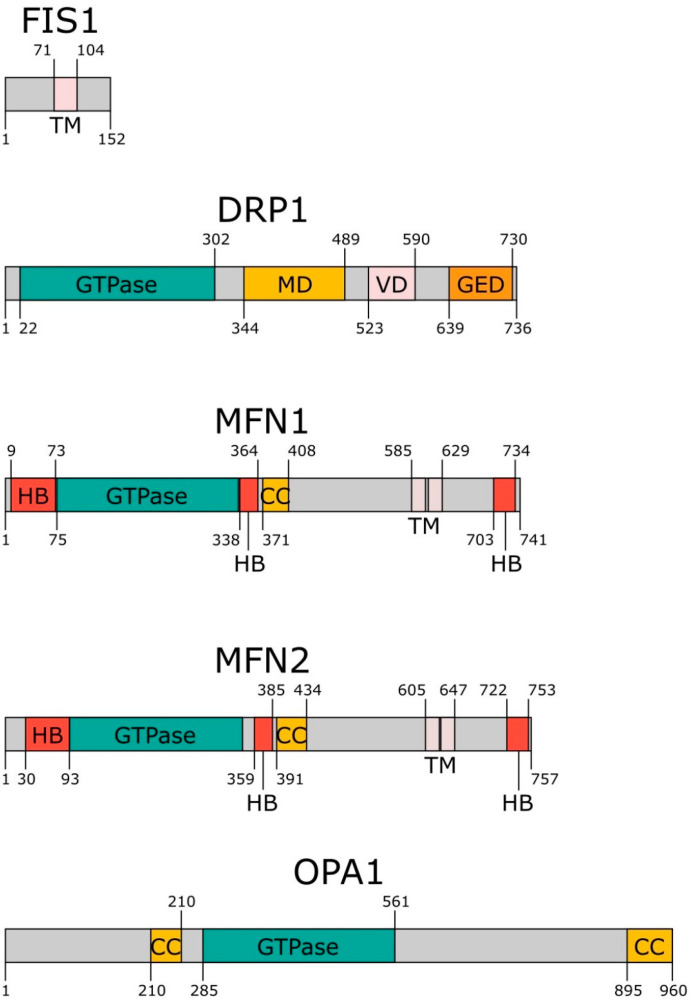
Domain structure of mitochondrial proteins: FIS1, DRP1, MFN1, MFN2, and OPA1. Abbreviations: CC—coiled-coil domain; GED—GTPase effector domain; GTPase—GTPase domain; HB—helix bundle domain; MD—middle domain; TM—transmembrane domain; VD—variable domain.

**Figure 2 ijms-24-13033-f002:**
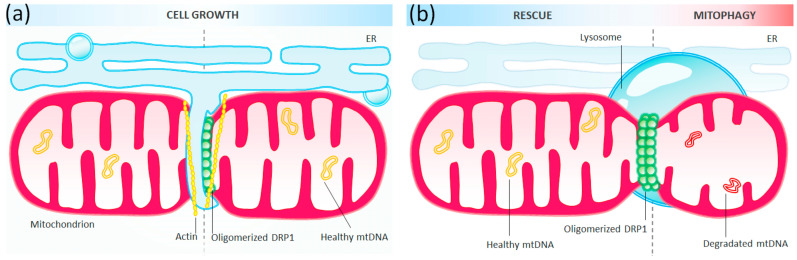
Mitochondrial Fission: (**a**) The Midzone fission process involves essential components such as the endoplasmic reticulum (ER) and actin, which initiate constriction before DRP1 binds to the outer mitochondrial membrane through adaptor proteins. This binding facilitates the scission process, leading to the division of the mitochondria. (**b**) In the case of peripheral fission, the process relies on an anchor point that bind to another organelle, the lysosome.

**Figure 3 ijms-24-13033-f003:**
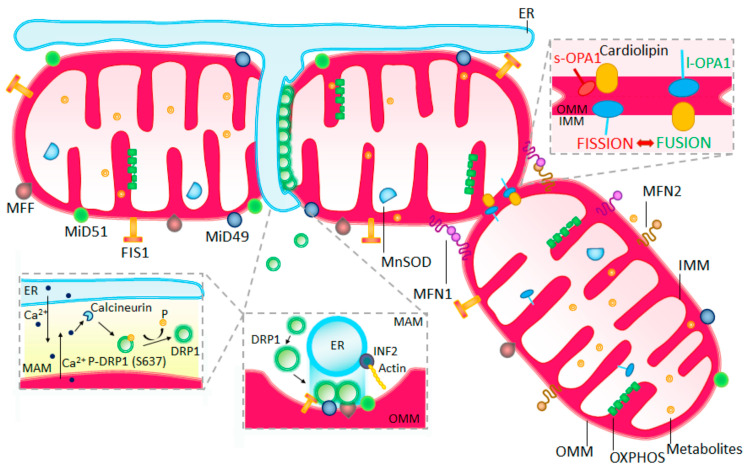
Mitochondrial fusion and fission, interaction between proteins.

## Data Availability

Not applicable.

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
