# Peer review of "Mitochondrial Dynamics in Neurodegenerative Diseases: Unraveling the Role of Fusion and Fission Processes"

_ijms, 2023, doi:10.3390/ijms241713033_

Round 1
Reviewer 1 Report
The review of H.Grel et al aims to identify the role of mitochondrial dynamics in neurodegenerative diseases (NDs). The authors try to convince that affecting fission-fusion processes in mitochondria can provide a principally new approach to the treatment of different NDs. The mechanisms of fission-fusion processes and the proteins involved are briefly described but, according to the authors themselves, full decoding is still far away and unilateral intervention through the influence on individual proteins is still premature. To improve the quality of mitochondria by influencing the dynamics and mitophagy looks more understandable. The authors mention this phenomenon very briefly, but a more detailed discussion is necessary. However, in terms of the volume and significance of the collected data, the review can be published in ijms.
Reviewer 2 Report
The review of Hubert Grel and coauthors “Mitochondrial Dynamics in Neurodegenerative Diseases: Unraveling the Role of Fusion and Fission Processes” describes the processes of fusion and fission in mitochondria in the context of neurodegenerative diseases. The authors discuss the complex interactions that underlie the development of neurodegenerative diseases. The aim of the review is to find targeted interventions and therapeutic strategies that will ultimately improve the quality of life of patients with neurodegenerative diseases. The paper is well presented and well written. The review consists of 9 parts. Each part is informative.
There is no doubt that this review should be published in the journal.
There are some comments on submitting the article.
1) Line 84… APOE gene … I think that it is necessary to decipher the abbreviation, or since this term is used once, write it in full as apolipoprotein E.
2) Lines 263, 355… remove italics [3] and [95].
3) Line 427… Ca2+ replace with Ca2+.
Minor editing of English language required
